# Outbreak of cutaneous leishmaniasis amongst militia members in a non-endemic district under conflict in the lowlands of Somali Region caused by Leishmania tropica, Eastern Ethiopia

Adugna Abera[1,2☯*], Henok Tadesse[3☯], Dereje Beyene[2], Desalegn Geleta[3], Mahlet Belachew[1], Ebise Abose Djirata[3], Solomon Kinde[1], Hailemariam Difabachew[1], Tesfahun Bishaw[4], Mussie Abdosh Hassen[5], Abdulahi Gire[6], Tariku Mulatu Bore[6], Binyam Mohammedbirhan Berhe[6], Medhanye Habtetsion[3], Zalalam Olani Tugga[3], Endawoke Eyelachew[3], Worku Birhanu Sefer[7], Kaoutar Choukri[8], Jasmine Coppens[9], Gemechu Tadese[10], Kebron Haile[4], Henock Bekele[11], Zeyede Kebede[11], Gert van der Auwera[8], Fikre Seife[4], Melkamu Abte[12], Getachew Tollera[12], Mesay Hailu[12], Jean-Claude Dujardin[8], Johan van Griensven[13], Dawit Wolday[12,14,15], Wendemagegn Embiale[16,17], Myrthe Pareyn[13‡], Geremew Tasew[1‡]

1 Malaria and Neglected Tropical Diseases Research Directorate, Ethiopian Public Health Institute, Addis Ababa, Ethiopia, 2 Department of Microbial Sciences and Genetics, College of Natural and Computational Sciences, Addis Ababa University, Addis Ababa, Ethiopia, 3 Center for Public Health Emergency Management, Ethiopia Public Health Institute, Addis Ababa, Ethiopia, 4 Federal Ministry of Health Ethiopia, Addis Ababa, Ethiopia, 5 Disease Prevention and Control Directorate, Somali Regional Health Bureau, Jigjiga, Ethiopia, 6 College of Health Sciences, Jigjiga University, Jigjiga, Ethiopia, 7 Animal Health Institute, Sebeta, Ethiopia, 8 Unit of Molecular Parasitology, Institute of Tropical Medicine, Antwerp, Belgium, 9 Clinical Reference Laboratory, Institute of Tropical Medicine, Antwerp, Belgium, 10 Department of Infectious Diseases Research Directorate, Ethiopian Public Health Institute, Addis Ababa, Ethiopia, 11 World Health Organization Country Office, Addis Ababa, Ethiopia, 12 Ethiopia Public Health Institute, Addis Ababa, Ethiopia, 13 Unit of Neglected Tropical Diseases, Institute of Tropical Medicine, Antwerp, Belgium, 14 Department of Biochemistry and Biomedical Sciences, Michael G. DeGroote Institute for Infectious Diseases Research (IIDR), Hamilton, Ontario, Canada, 15 McMaster Immunology Research Center (MIRC), Health Sciences, McMaster University, Hamilton, Ontario, Canada, 16 Department of Dermato-venereology, College of Medicine and Health Sciences, Bahir Dar University, Bahir Dar, Ethiopia, 17 Collaborative Research and Training Center for Neglected Tropical Diseases, Arba Minch University, Arba Minch, Ethiopia

☯ These authors contributed equally to this work.

‡ These authors are co-senior authors.

* adugnabe@yahoo.com

## Abstract

### Background

Cutaneous leishmaniasis (CL) in Ethiopia has typically been linked to high-altitude regions but has recently emerged at an unusually low altitude of 500 meters in the Somali Region, raising public health concerns. Cutaneous leishmaniasis has not been previously identified in the region. There is a conflict in the starting area and only militias have been infected with very serious lesions.

**Data availability statement:** All relevant data are available in the manuscript and its supporting information files.

**Funding:** J.vG and G.T. received funding from the Directorate-General for Development Cooperation and Humanitarian Aid under the Framework Agreement 5 collaboration between the Institute of Tropical Medicine (Antwerp, Belgium) and Ethiopian Public Health Institute (Addis Ababa, Ethiopia). The funders had no role in study design, data collection and analysis, decision to publish, or preparation of the manuscript.

**Competing interests:** The authors have declared that no competing interests exist.

## Methodology/Principal findings

Routine clinical and socio-demographic information was extracted from the patient chart using a case report form. Additionally, clinical and laboratory data were obtained from 30 patients suspected for CL. Skin scraping and fine needle aspirates were collected from the raised edges, nodular and centre of the lesions followed by DNA extraction using the DNeasy Blood and Tissue kit. There were a total of 1050 CL patients recruited, all of them were male militia members, immunologically naïve and displaced into a conflict area with a likely sylvatic transmission cycle. We identified *Leishmania tropica* as the causative species, challenging the previous assumption that *L. aethiopica* was the primary agent of CL in Ethiopia. Notably, over 77% of patients had more than 10 lesions, a presentation atypical for *L. tropica* elsewhere. *Phlebotomus orientalis* and *P. sergenti*, vectors for visceral leishmaniasis and CL in North Africa respectively, were found in the outbreak area.

## Conclusions/Significance

Further research is needed to explore the eco-epidemiology of the outbreak and patient's treatment responses. Insights will help develop management strategies to control this newly emerging form of CL, prevent its spread to other regions and hybridization with *Leishmania* strains causing VL endemic in the area.

### Author summary

Cutaneous leishmaniasis (CL) is a significant public health problem in Ethiopia and is spread by female sandflies. The causative agent is *Leishmania aethiopica*, and the reservoir hosts are rock and tree hyraxes. After being bitten by a sand fly, distinct skin lesions typically develop within 6 weeks of infection. The ulcers can persist for varying durations and heal slowly and causing ugly scarring. This disease is mainly found in the highland areas of the country. However, there have been recent changes in the transmission patterns of CL in Ethiopia, with the disease now being identified in lowland areas and involving different hosts and vectors. In recent reports, there have been findings of hybrid parasites resulting from *L. donovani*/*L. aethiopica* and *L. donovani*/*L. tropica* species. Additionally, cases of CL caused by *L. donovani* have been documented. A high number of cases, predominantly among male militia members, were reported following the recent outbreak of CL in the Somali region of Ethiopia. Our study has identified the infecting Leishmania species as *L. tropica*, causing atypical skin lesions. This study sheds light on the emergence of *L. tropica* as a new Leishmania species, responsible for causing skin lesions in the lowlands of Ethiopia.

## Introduction

Leishmaniasis is a neglected tropical disease (NTD) caused by *Leishmania* protozoa. Depending on the infecting parasite species, it can manifest as either cutaneous (CL) or visceral (VL) leishmaniasis, which both pose an important public health problem in Ethiopia [1].

Visceral leishmaniasis, caused by *Leishmania donovani,* is predominantly prevalent in the lowlands in the northwest and southern Ethiopia. The primary vectors facilitating anthroponotic transmission in these regions are *Phlebotomus orientalis* and *P. martini* [2]. Annually, treatment centers report 2,000–4,000 VL cases [1].

In contrast, CL primarily occurs at altitudes ranging from 1650 to 2700m, predominantly on the mountain slopes of the Ethiopian Great Rift Valley. The region extends from northwest to southwest and south to east Ethiopia [3]. The vectors responsible for CL, *P. longipes* and *P. pedifer* [4,5] thrive in cool, humid climates and rocky terrains. Transmission of CL is zoonotic, with hyraxes (*Heterohyrax brucei* and *Procavia capensis*) serving as main animal reservoirs [4]. Annually, there are an estimated 20,000–50,000 CL cases in Ethiopia [1]. Clinically, CL is categorized in three forms: localized CL (LCL), mucocutaneous leishmaniasis (MCL), and diffuse CL (DCL) [6,7]. Most commonly, CL manifests as localized plaques or nodules on exposed body parts, although involvement of the mucosa also occurs regularly [8]. With no standard treatment for CL in Ethiopia, common practice involves either weekly intralesional or daily systemic injections of sodium stibogluconate for a month [9].

While *Leishmania aethiopica* is the primary causative agent of CL in Ethiopia, evidence suggests potential involvement of other species. *Leishmania tropica* and *L. major*, the CL-causing species in northern Africa, have been found in Ethiopian animals (*e.g.,* bats, rodents) [10,11] and sand flies [12,13] which could potentially be disease reservoirs and vectors, respectively. Most clinical studies have identified *L. aethiopica* as the species responsible for CL [14–16], with two exceptions. A single CL case caused by *L. tropica* was reported in Awash valley in eastern Ethiopia in 2006 [14]. Furthermore, in a recent study we demonstrated that *L. donovani* was the causative species for four CL patients in the northwest of Ethiopia [15], two of whom were from VL-endemic lowland areas. Additionally, a laboratory study examining the genomic diversity of *L. aethiopica* cultures revealed two interspecific hybrids, one *L. aethiopica*/*L. tropica* hybrid, and one *L. aethiopica*/*L. donovani* hybrid [16]. These findings underscore the complexity of CL in Ethiopia and the need for further research on the *Leishmania* species involved.

In August 2023, the Ethiopian Ministry of Health (MoH) and Ethiopian Public Health Institute (EPHI) received reports of a significant number of extremely severe CL cases in a lowland area of Duunyar district in the eastern Somali Region in Ethiopia, which is under conflict. The area has no prior history of CL cases, although surrounding areas do sporadically report VL cases. In response, EPHI's Public Health Emergency Management (PHEM) unit conducted a rapid outbreak investigation to better understand the causative species, type of sand flies presents and characterize the patients. This information will be used to manage the outbreak, inform policy makers and provide a foundation for future research in this newly identified CL focus.

## Materials and methods

### Ethics statement

The study was approved by the Institutional Review Boards (IRBs) of the Ethiopian Public Health Institute (EPHI-IRB-554-2024), Institute of Tropical Medicine (ITM) Antwerp (ref 1745/24) and the University Hospital of Antwerp (ref 6229). We obtained formal verbal informed consent from all participants aged 18 years and older. For individuals aged <17 years, we obtained verbal informed consent from a parent or guardian. All study participants consented to use their de-identified pictures where applicable for the study. All positive cases eligible were treated at Sitti Primary Hospital and Sheik Hassen Yabare Comprehensive Specialized Hospital. A material transfer agreement (MTA) was developed between Ethiopian Public Health Institute and ITM for transfer of DNA samples for species typing.

## Study setting

The Duunyar district is situated in the Sitti Zone of the Somali Region in eastern Ethiopia, bordering the Afar region (Fig 1). Badhi Wayne is the main town in this newly formed district. Occasionally, VL cases are reported from this area.

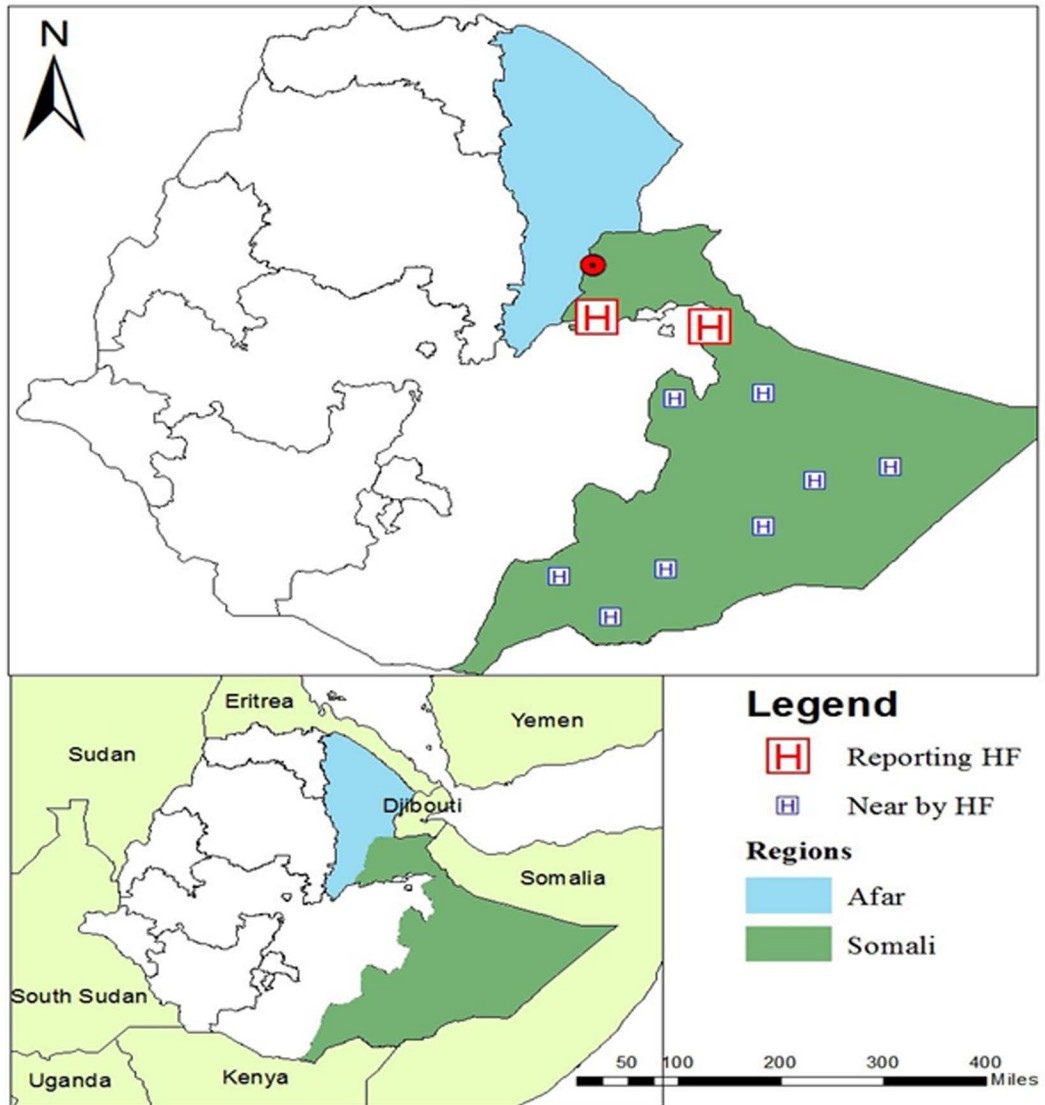

**Fig 1. Map indicating the location of the outbreak site and healthcare facilities in Somali Region in Ethiopia (UNOCHA, 2021).** Map of Ethiopia showing neighboring countries, with the Afar region highlighted in blue and the Somali region in green. The Duunyar district with Badhi Wayne town, where the outbreak occurred, indicated by a red dot, is situated at the border between the two regions, on the Somali side. Duunyar Health Center is located here as well. Sitti Primary Hospital and Jigjiga University Sheik Hassen Yabare Comprehensive Specialized Hospital are situated in the northern part of Somali Region (Red "H"). Other healthcare facilities (including Karamara General Hospital, Shinile Primary Hospital, Gode General Hospital, Fik Primary Hospital, Filtu Primary Hospital, Hargele General Hospital, Warder Primary Hospital, Dawa Primary Hospital and Dollo Ado Primary Hospital) from which CL cases are reported in Somali region are further from the outbreak site and indicated by a blue "H".

The maps were built using the free and open source QGIS software version 3.36.3 (QGIS Development Team (2024). QGIS Geographic Information System, version 3.36.3. Open-Source Geospatial Foundation Project. https://qgis.org) and shapefiles were obtained from the free and open-source site. https://data.humdata.org/.

Duunyar is situated at an altitude of about 400–600 m. Daytime temperatures often exceed 30°C and can soar above 40°C during the hottest months. The area experiences less than 300 mm rainfall per year, which often occurs in intense bursts, leading to occasional flooding. Long periods of drought are common, contributing to harsh living conditions. The landscape is primarily arid and semi-arid, characterized by sparse vegetation and rocky terrains (Fig 2A).

The area is highly marginalized, remote and considered one of the least developed in the country in terms of infrastructure. Previously, Duunyar was only sporadically inhabited with nomadic populations (Fig 2B) moving frequently in search of water and grazing land for livestock, until conflicts began. This unrest has been driven by local clan conflicts that extend across domestic borders. Camps with military police men are therefore currently occupying the region. The conflict area is located approximately 20 km (around a 4-hour walk) north of Badhi Wayne, where military camps are established. The area looks ecologically similar to Badhi Wayne.

Healthcare facilities in Duunyar are extremely limited, with only a few primary healthcare centers that are often under-resourced and understaffed. The region's remote and conflict-prone nature further complicates healthcare delivery, as poor infrastructure hampers the transport of medical supplies and patients. The Duunyar Health Center, located in Badhi Wayne town, offers limited services and is primarily capable of identifying suspected CL patients. For treatment, patients are referred to hospitals in larger towns or cities. Historically, Jigjiga University Sheik Hassen Yabare Comprehensive Specialized Hospital, located 300 km from Duunyar, was the only relatively nearby facility equipped to diagnose and treat VL and CL. In response to the outbreak, diagnosis and treatment services were also made available at Sitti Primary Hospital in Gota Biki town, approximately 150 km from Duunyar, through the efforts of EPHI, the regional health bureau, and Médecins Sans Frontières (MSF).

### Line list data collection

The line list of suspected CL patients was collected from 11 healthcare facilities in the wide areas of Somali region ("H" in Fig 1) from March until December 2023. From August onwards, structured patient formats were introduced at all the healthcare facilities for systematic data collection. Data included were socio-demographics, occupation, travel history to endemic areas, date of onset of symptoms and the number and site of lesions.

### Study population of the outbreak investigation

Patient data and sample collection were performed at Sitti Primary Hospital and Duunyar Health Center by EPHI's PHEM team; and at Sheik Hassen Yabare Comprehensive Specialized Hospital by the Ministry of Health (MoH) *Leishmania*

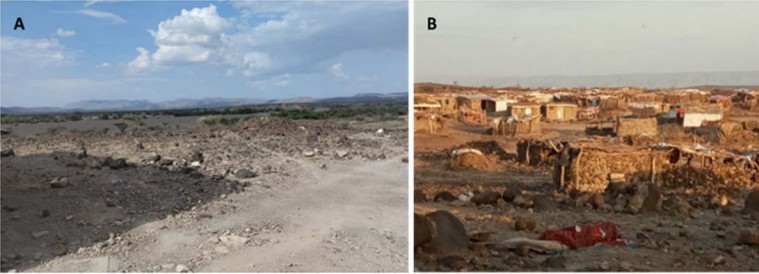

**Fig 2. Pictures of the landscape nearby the outbreak site in Duunyar district, Ethiopia. A.** Dry weather causing a semi-arid to arid landscape, characterized by rocky terrains. **B.** Settlements of nomadic populations in the district. Pictures: Hailemariam Difabachew.

division and World Health Organization (WHO) country office teams. Patients suspected of CL (all adults) presenting during the visit of the study teams were included after providing oral informed consent. Suspected cases were defined by the presence of clinical signs of CL as determined by the treating physician, without requiring parasitological confirmation.

## Outbreak investigation data collection

Routine sociodemographic (age, sex, occupation) information was extracted from the patient charts using a case report form (CRF) only for the patients recruited by the PHEM team. Pictures of CL lesions were taken and immediately edited to ensure patients could not be recognized. Based on the pictures collected during the outbreak investigation, an experienced dermatologist classified patients as having LCL, MCL or DCL and characterized the lesions.

## Sample collection

Two types of samples were collected: a skin scraping at Sitti Primary Hospital and Duunyar Health Center and fine needle aspirates (FNA) at the Sheik Hassen Yabare Comprehensive Specialized Hospital.

Skin scrapings were performed by scraping off tissue with a scalpel either from the raised edge, nodular part or center of the lesion and stored in Locke's solution prepared at EPHI in cold-chain. Additionally, a skin scraping was collected for Giemsa staining and microscopic examination to detect amastigotes. Amastigotes were quantified according to Chulay and Bryceson's method [17].

FNA samples were collected by experienced pathologists by injecting 0.1 mL of sterile 0.9% NaCl with a 21-gauge needle into the lesion's active border through unharmed skin. A gentle suction was performed when retracting the needle and the aspirate was transferred in Locke's solution.

Both sample types stored in Locke's solution were transported to EPHI in cold-chain and stored at -80°C until laboratory analyses.

## *Leishmania* detection and species typing

DNA was isolated from the FNA using the DNeasy Blood and Tissue Kit (Qiagen) and from the skin slits using the Geneius Micro gDNA Extraction kit (Geneaid) according to the manufacturer's instructions.

Samples were tested for *Leishmania* infection using a qPCR targeting the internal transcribed spacer-1 (ITS-1) gene, as previously described by Talmi-Frank *et al.* [18]. The assay was run including *L. aethiopica* and *L. donovani* (*L. tropica* not available) positive controls and non-template controls (nuclease free water) on a QuantStudio 5 (Applied Biosystems) instrument at EPHI.

qPCR-positive DNA samples were sent to the Institute of Tropical Medicine Antwerp for confirmatory species typing. Samples were subjected to a conventional PCR targeting the 1245 bp F-fragment of the heat-shock protein 70 (HSP70) gene, as described by Van der Auwera et al. [19,20], and visualized on a 1.5% agarose gel. If no band was observed, a nested PCR was conducted on 1 µL PCR product to amplify the overlapping 522 bp N fragment and 723 bp T fragment of the HSP70 gene separately. The quality of the amplicons was checked on a TapeStation instrument (Agilent) according to the manufacturer's instructions.

## Sandfly collection

A brief entomological assessment was performed in and around Badhi Wayne town. Two CDC light traps (John W. Hock Company, USA) and ten sticky traps (white polypropylene sheets coated with castor oil) were employed during four nights to capture sandflies. Traps were placed in peri-domestic habitats outdoors at dusk and collected at dawn. Traps were placed in stone piles, rodent burrows, cracked soil and animal enclosures. Captured sandflies were preserved in 97% ethanol and transported to EPHI. Specimens were sorted according to sex and genus (*Phlebotomus* and *Sergentomyia*).

The last three segments of the abdomen and head were dissected and mounted in Hoyer's medium for morphological species identification using relevant taxonomic keys [21–25].

## Results

### Epidemiological data from the line lists

A total of 1050 patients were clinically and/or parasitologically diagnosed with CL from March to December 2023. Their reported day of onset of symptoms is presented in Fig 3. The number of cases gradually increased from March onwards, peaking in July with over 700 patients in one month. After a sudden drop in cases in August, the number rose again, reaching another peak of approximately 200 cases in October. The attack rate of the outbreak could not be calculated because the number of involved combatants is classified information.

All patients were male regional militia members who reported deployment to the regional border conflict between Afar and Somali within the year preceding the onset of their symptoms (Table 1). The median age of the patients was 24 years (IQR 21–28), with 90.5% under the age of 35, 7.0% between 35 and 45 and the remaining 2.6% were over 45 years old.

Almost all cases (99.1%) presented with more than three lesions on their body, with 77.0% having even more than 10 lesions at time of presentation. Most patients (97.9%) had lesions on both their face and limbs. Only 1.9% had lesions on their limbs, and 0.2% had lesions solely on their faces.

### Socio-demographic and clinical characteristics of the CL patients from the outbreak investigation

The PHEM team recruited 15 patients from Duunyar Health Center and Sitti Primary Hospital, while the WHO and MoH teams collected data from 15 and 6 patients respectively, from the Sheik Hassen Yabare Comprehensive Specialized Hospital. Similar to the line list data, all patients were male, regional militia members with a median age of 28 (IQR 22–35). Pictures were collected from 22 patients, of which the ones with best quality and different characteristics are presented

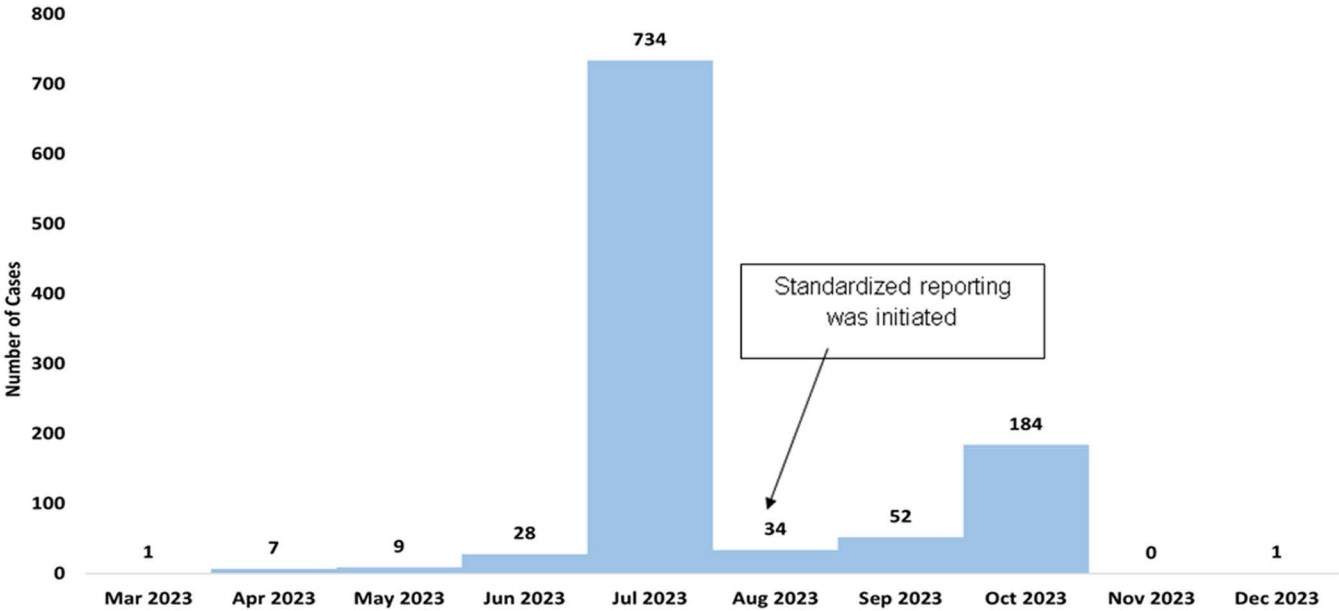

**Fig 3. Epidemiological curve of cutaneous leishmaniasis outbreak in Somali Region showing the number of cases diagnosed with their date of onset of symptoms.** Data are compiled from 11 reporting healthcare facilities in the Somali Region. A standardized reporting system was introduced in August 2023.

**Table 1. Socio-demographic and lesion characteristics of cutaneous leishmaniasis cases from the line list of 11 reporting hospitals in Somali region.**

| Characteristics | | Number | Proportion |
|---|---|---|---|
| **Sex** | | | |
| | Male | 1050 | 100% |
| **Age (years)** | Median 24 (IQR 21–28) | | |
| | 15-24 | 552 | 52.6% |
| | 25-34 | 398 | 37.9% |
| | 35-44 | 73 | 7.0% |
| | ≥45 | 27 | 2.6% |
| **Number of lesions** | | | |
| | 1-3 | 9 | 0.9% |
| | 4-6 | 98 | 9.3% |
| | 7-9 | 134 | 12.8% |
| | ≥10 | 809 | 77.0% |
| **Site of lesion** | | | |
| | Face and Limbs | 1028 | 97.9% |
| | Face | 20 | 1.9% |
| | Limbs | 2 | 0.2% |

IQR: interquartile ranges. Total number of clinically diagnosed CL patients is 1050, which was used as denominator for calculating the proportions.

(Figs 4 and 5). Patients presented with a variety of clinical forms and morphological presentations. However, each patient presented with monomorphic lesions (either papular, erosive, ulcerative or psoriasiform). The open lesions, including erosive or ulcerative lesions, were wet and mostly had a yellowish crust. One fourth of the patients had mucosal ulceration or crusts. Lesions often appeared clustered on specific body parts.

### Laboratory results

Microscopic results were available for the 15 patients recruited by the PHEM team, of which 8 were positive by microscopy: 3 with +6, 1 with +5, 1 with +4, 1 with +3 and 2 with +2 parasite load scores (see S1 Fig).

PCR targeting ITS-1 identified 19 out of 36 samples as positive for *Leishmania*. Species identification was made by HSP70 amplicon sequencing. A chromatogram of the HSP70 gene was obtained for 18 samples, although poor sequence quality led to several ambiguities in five of them. Nevertheless, these ambiguities did not result in any differences from the other obtained sequences. Consequently, the species for all 18 samples was determined to be *L. tropica*. These sequences clustered together with *L. tropica* and separately from *L. aethiopica, L. major* and *L. donovani* (Fig 6).

### Entomological survey

During the four trapping nights in peridomestic habitats, 25 sandflies were trapped of which 17 belonged to the *Phlebotomus* genus. Fifteen (60%) were identified as *P. orientalis*, three males and 12 females. Two male sand flies (8%) were *P. sergenti* and the remaining 8 were *Sergentomyia* (Table 2).

During sand fly trapping, an abundance of rodents and hyraxes were observed by the study team in the area, particularly around the rocky peridomestic sites.

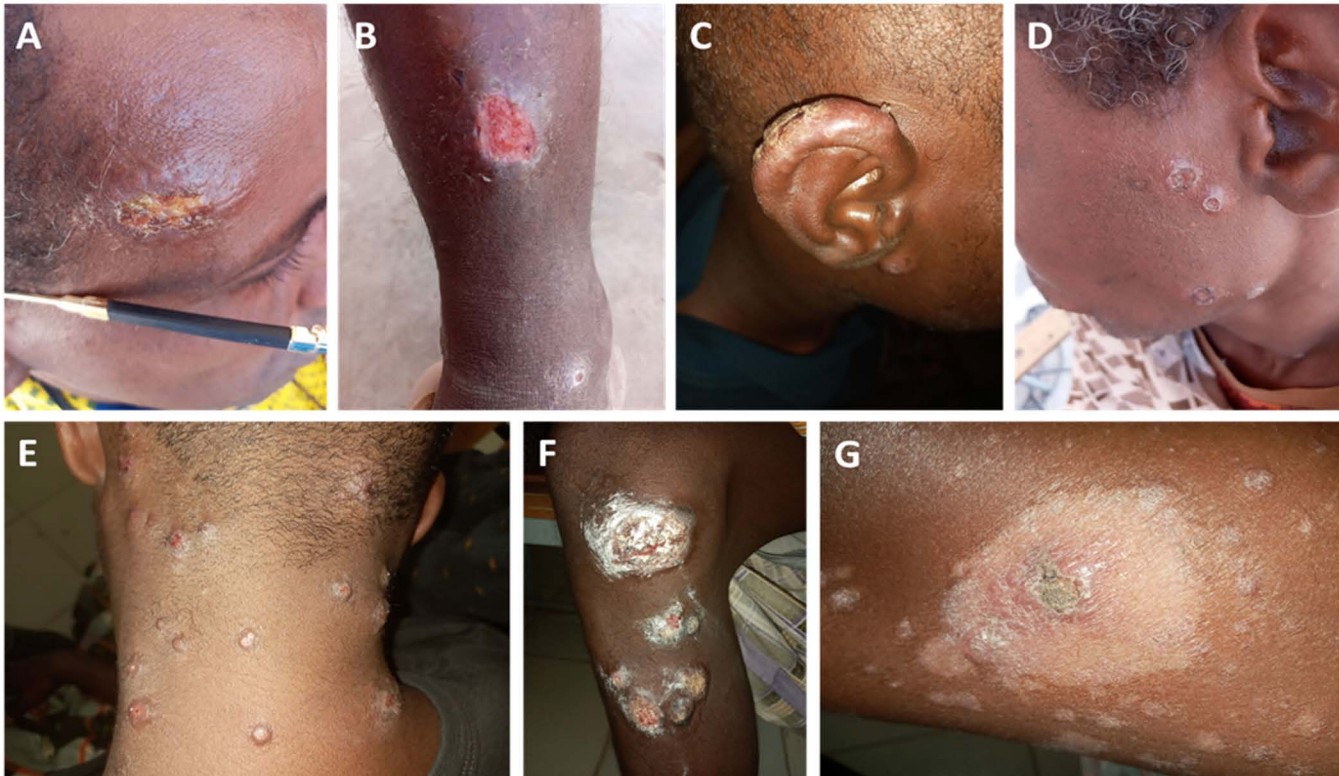

**Fig 4. Cutaneous leishmaniasis in Somali region, Ethiopia.** A: Solitary infiltrative nodule with central erosion and a yellowish crust on the temporal area. B: Ulcer on the shin of the leg with a clean granulated base and a rim of raised borders. C: Edematous swelling of the helix of the ear, featuring an ulcer with a dry crust. D: Multiple dry, infiltrative papules with crater-like central dimpling on the left side of the face. E: More than 17 skin-colored papules scattered on the nape of the neck, occipital part of the scalp, and posterior auricular area. F: Multiple discrete infiltrative psoriasiform plaques with grey crust/scaling and areas of erosion on the leg below the knee. G: Infiltrative plaque with central crusting, surrounded by a peripheral hypopigmented patch with fine scaling and multiple satellite papulosquamous lesions on the leg. Pictures: Hailemariam Difabachew.

## Discussion

Cutaneous leishmaniasis in Ethiopia has historically been associated with higher-altitude regions [26,27]. However, this recent outbreak at an unusually low altitude of approximately 500 meters in the Somali Region raises concerns in the country. Previously, CL cases had not been reported in this area or sporadically but remained unreported. Cases were all militia members deployed to the battlefield and appear clinically different from the rest of the country, making this outbreak notable from both geographical and clinical perspectives. It is crucial to investigate this emerging transmission focus to better understand the dynamics of the outbreak and prevent further cases, potential spread to other areas or hybridization with VL-causing *Leishmania* species as already reported in Ethiopia [16].

Based on the hospital records, we show that the outbreak peaked in July 2023, with nearly 800 reported cases. This surge coincided with the escalation of conflict in May and June 2023, when militia forces—of whom most were presumably immunologically naïve to *Leishmania*—were deployed to the region. The timing aligns with CL's incubation period, which ranges from two weeks to several months. Following the peak, reported cases temporarily declined in August and September, likely due to policies requiring soldiers with mild or asymptomatic infections to remain in the field. A second surge occurred in October, following a temporary de-escalation of the conflict and the withdrawal of militia forces. However, as the conflict has flared up again since then, further cases are likely as long as the situation persists. This poses an ongoing

PLOS Neglected Tropical Diseases

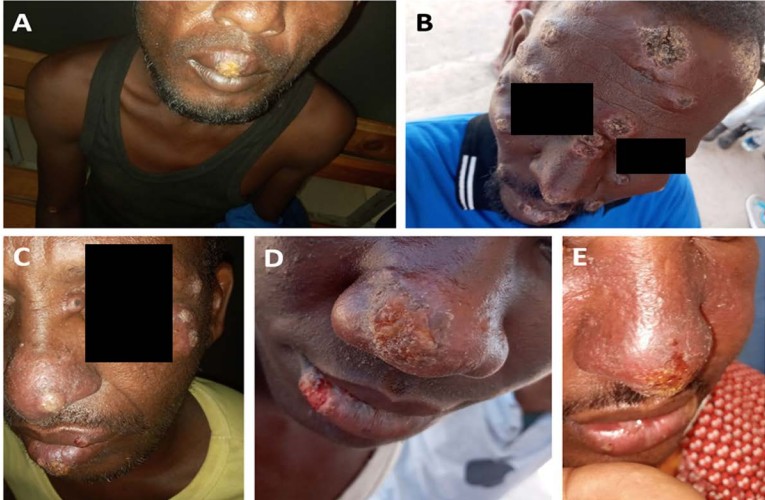

**Fig 5. Mucocutaneous leishmaniasis patients in Somali Region, Ethiopia.** A: Solitary ulcer with a yellowish thick crust on the upper lip mucosa. B: Multiple (more than 18) discrete nodular lesions with central dark grey dry thick crust and infiltrative margins scattered on the face and lip. C: Multiple papular lesions scattered on the left temporal region, nasal bridge, and tip of the nose, along with yellowish crusted ulcers on the upper and lower lips. D: Erosive lesion with a yellowish crust on the right tip of the nose and lower lip. F: Infiltrative swelling of nose cheek and lip with erythema. Erosive ulcer on tip of the nose. Pictures: Hailemariam Difabachew.

risk, not only for those in the conflict zone but also for other regions in Ethiopia, if infectious militia travel to new areas where competent sand fly vectors may be present.

We identified *Leishmania tropica* as the causative agent of this outbreak, which is relatively rare in Ethiopia; only one case had previously been documented two decades ago in the Awash Valley [14] and possible *L. aethiopica*/*L. tropica* hybrid parasites [16], but the altitude and ecology of that area, nor clinical presentation of the patient were described in the report. A parallel study is in progress, in which we report whole genome sequences of the Somali *L. tropica* samples, their homogeneity and relatedness with other *L. tropica* collected worldwide.

The clinical presentation of patients identified in this outbreak differs markedly from what is typically observed in Ethiopian CL cases caused by *L. aethiopica*. In the Ethiopian highlands, CL lesions caused by *L. aethiopica* are generally dry, solitary (although some patients may present with a few lesions due to multiple sand fly bites or DCL), with crusting and localized erythema or dark brown discoloration [6–8]. In contrast, 99% of patients in this outbreak presented with more than three lesions, and 77% had over ten lesions, primarily wet and monomorphic in nature. This differs also significantly from *L. tropica* infections in other regions, such as Kenya and North Africa, where single, dry, chronic lesions are more common [28].

The presentation of multiple, clustered lesions could be attributed to various factors. One possibility is that a different strain of *L. tropica*, potentially producing virulence factors that promote clustered lesion formation, is involved. An epidemic increase in cases with more aggressive pathology (although not characterized) has been reported from Sudan. Although this remained uninvestigated, authors hypothesized it could be caused by *L. tropica*, a novel emergent *Leishmania* species or a new strain of *L. major*, which generally causes CL in Sudan [29]. Alternatively, highly localized immune responses at the bite sites could lead to the formation of multiple lesions in proximity. Sand fly behavior under extreme environmental conditions, such as those in the Somali Region where temperatures can reach up to 45°C, may also be contributing to the observed presentation. Sand flies are known to probe multiple times before feeding, and environmental stressors like heat may shorten their feeding times, causing them to bite repeatedly in a small area.

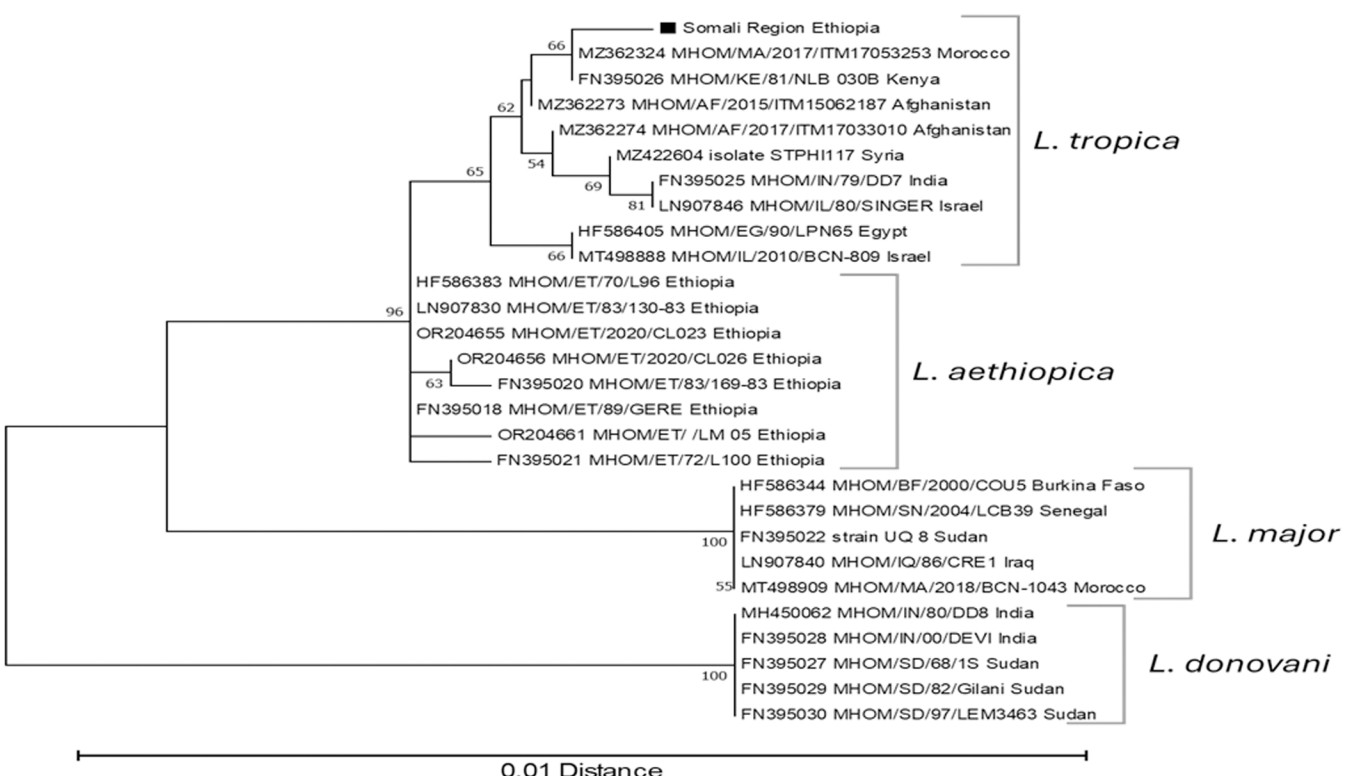

**Fig 6. Dendrogram of HSP70 sequences from Leishmania parasites causing cutaneous leishmaniasis in Somali Region, Ethiopia.** The sequence identified as "Somali Region Ethiopia" is representative of the parasites S02/ S05/ S06/ S07/ S11/ SH02/ SH03/ WHO1/ WHO6/ WHO8/ WHO9a/ WHO11/ WHO12. For parasites S04/ WHO4/ WHO9b/ WHO13/ WHO15 several ambiguities were seen from bad sequence quality, but none of these showed a difference from the other Somali Region Ethiopia sequences. Obtained sequences are compared to representatives of Old World Leishmania species. Sequences were recovered from Genbank, and are identified by their accession number, followed by strain identification and country of origin as recorded in the GenBank sequence entry. Bootstrap values from 2000 replicates are shown in percentages at the internodes when higher than 50%. The distance scale is shown at the bottom. The root was placed on the branch leading to the L. donovani species.

**Table 2. Morphological species identification of sandflies captured in and around Badhi Wayne town, Somali Region.**

| Species | Male | Female | Total | Proportion |
|---|---|---|---|---|
| *Phlebotomus orientalis* | 3 | 12 | 15 | 60% |
| *Phlebotomus sergenti* | 2 | 0 | 2 | 8% |
| *Sergentomyia squamiplueris* | 1 | 4 | 5 | 20% |
| *Sergentomyia clydei* | 0 | 3 | 3 | 12% |

The presence of *Phlebotomus orientalis* and other sand fly species such as *P. bergeroti*, *P. saevus*, and *P. alexandri* has been documented before in the lower Awash Valley, with similar environments as the current outbreak site [13]. However, *P. sergenti*, the primary vector for *L. tropica* in North Africa, was not found in these lower altitudes in previous studies in Ethiopia [28], but was detected at our outbreak site. It is plausible that *P. sergenti* could also be playing a role in transmitting *L. tropica* in this region of Ethiopia. As VL cases have been reported from the area, there is potential for inter-species hybridization of co-existing *L. tropica* and *L. donovani* in the sand fly midgut, if *P. orientalis* appears a permissive vector.

During our entomological survey, only a limited number of sand flies was captured (due to safety concerns) and the study site was located approximately 20 kilometers from the conflict area, although in an ecologically similar area. Therefore, our results may not reflect the full diversity of sand fly species present in the area. More comprehensive entomological surveys are needed to identify which species are responsible for transmitting the parasite in this outbreak.

Additionally, hyraxes and rodents, which were observed in the outbreak area, could serve as animal reservoirs for the parasite. In Ethiopia, hyraxes are known reservoirs for *L. aethiopica*, while in Kenya, they have been implicated in the transmission of *L. tropica* [4,30–32]. Similarly, rodents such as *Gerbillus nanus* and *Acomys* spp. have been found infected with *L. tropica* in the Awash Valley [10], suggesting that a sylvatic transmission cycle involving these animals might exist in this region. This could explain how transmission persisted in the area, with sporadic spillover into human populations, for instance when the conflict led to the deployment of militia to the area.

The emergence of this new CL focus has significant public health implications, especially given the potential for the disease to spread to other regions with susceptible populations and vectors if infectious militia members would migrate without proper treatment. To prevent further outbreaks, conducting epidemiological studies to identify infection sources, vectors, and potential animal reservoirs is crucial and patient care should be optimized. Moreover, it will be used to inform development of effective control measures and prevent further spread.

Species typing should be a priority in future research to better understand the epidemiology of CL in Ethiopia. Previously, it was thought that *L. aethiopica* was the sole cause of CL in the country, but recent studies, including this outbreak and research showing *L. donovani* causing CL, indicate that multiple species are involved [15]. Whole-genome sequencing in particular with methods allowing direct analysis of clinical samples [33], could help clarify strain variations, identify potential hybrids, and provide insights into transmission dynamics. This could also help inform treatment protocols and improve disease control strategies.

In conflict zones, conducting research is challenging, but following previously infected soldiers to identify patterns of infection and disease progression could provide valuable information. Additionally, comprehensive eco-epidemiological studies are necessary to identify the specific vectors and animal reservoirs involved in this outbreak. This would aid in developing more targeted control measures for both local populations and military personnel, who may be at high risk for future infections.

In response to the outbreak, the Ethiopian Ministry of Health has already adapted its treatment guidelines to those used by MSF for *L. tropica*. However, systematic documentation of treatment outcomes in this specific outbreak is needed to determine the most effective regimens for this population. Given the variability in treatment responses across different *Leishmania* species and strains, continuous monitoring of species involved is essential to ensure that patients receive the most appropriate care.

Furthermore, improving Ethiopia's preparedness for future CL outbreaks is vital. This includes ensuring that sufficient treatment supplies are available, especially given the potential for drug shortages in the country. Strengthening diagnostic capacity, treatment access, and vector control measures are also essential steps to mitigate the risk of further spread and outbreaks.

Finally, developing detailed guidelines for managing CL in this region is crucial. These guidelines should address clinical presentation, treatment protocols, and preventive measures. They will ensure that local health workers, militia members and nomadic populations can protect themselves and will effectively be managed. Infrastructure development in the affected region, along with enhanced healthcare access, will also be critical to reducing the impact of future outbreaks.

## Supporting information

**S1 Fig. Microscopy image of skin slit collected from a cutaneous leishmaniasis patient in Duunyar Health Center.** (TIF)

**S2 Fig. Results of PCR targeting ITS-1 followed by high-resolution melt curve analysis.** In red is the L. aethiopica positive control, in yellow L. donovani and in grey the samples' melting curves. Melting temperature (Tm) for L. donovani

is 81.8 and for L. aethiopica 83.4. Samples are all situated somewhere in between the Tm values of the two positive controls, where normally L. tropica lies with a Tm of around 82.4, but which was not available for the analysis done in Ethiopia.
(TIF)

**S1 Table. Raw data of high-resolution melt curves of ITS-1.** Fourteen out of 18 positive Leishmania samples provided a melting temperature (Tm). The Tm of samples did not match with that of the positive controls.
(DOCX)

## Acknowledgments

We thank members of outbreak investigation and control agencies in all regional, zonal, districts, and communities.

## Author contributions

**Conceptualization:** Adugna Abera, Dereje Beyene, Ebise Abose Djirata, Dawit Wolday, Myrthe Pareyn, Geremew Tasew.

**Data curation:** Adugna Abera, Henok Tadesse, Desalegn Geleta, Mahlet Belachew, Ebise Abose Djirata, Solomon Kinde, Hailemariam Difabachew, Tesfahun Bishaw, Mussie Abdosh Hassen, Abdulahi Gire, Tariku Mulatu Bore, Binyam Mohammedbirhan Berhe, Medhanye Habtetsion, Zalalam Olani Tugga, Endawoke Eyelachew, Worku Birhanu Sefer, Kaoutar Choukri, Jasmine Coppens, Kebron Haile, Henock Bekele, Zeyede Kebede, Gert van der Auwera, Myrthe Pareyn, Geremew Tasew.

**Formal analysis:** Adugna Abera, Henok Tadesse, Desalegn Geleta, Mahlet Belachew, Solomon Kinde, Gert van der Auwera, Johan van Griensven, Dawit Wolday, Wendemagegn Embiale, Myrthe Pareyn, Geremew Tasew.

**Funding acquisition:** Johan van Griensven, Geremew Tasew.

**Investigation:** Adugna Abera, Henok Tadesse, Dereje Beyene, Mahlet Belachew, Ebise Abose Djirata, Solomon Kinde, Hailemariam Difabachew, Tesfahun Bishaw, Mussie Abdosh Hassen, Abdulahi Gire, Tariku Mulatu Bore, Binyam Mohammedbirhan Berhe, Medhanye Habtetsion, Zalalam Olani Tugga, Endawoke Eyelachew, Worku Birhanu Sefer, Kaoutar Choukri, Gemechu Tadese, Kebron Haile, Henock Bekele, Zeyede Kebede, Gert van der Auwera, Fikre Seife, Jean-Claude Dujardin, Dawit Wolday, Geremew Tasew.

**Methodology:** Adugna Abera, Henok Tadesse, Dereje Beyene, Desalegn Geleta, Mahlet Belachew, Ebise Abose Djirata, Solomon Kinde, Hailemariam Difabachew, Tesfahun Bishaw, Mussie Abdosh Hassen, Abdulahi Gire, Tariku Mulatu Bore, Binyam Mohammedbirhan Berhe, Medhanye Habtetsion, Zalalam Olani Tugga, Endawoke Eyelachew, Worku Birhanu Sefer, Kaoutar Choukri, Jasmine Coppens, Kebron Haile, Henock Bekele, Zeyede Kebede, Gert van der Auwera, Fikre Seife, Jean-Claude Dujardin, Johan van Griensven, Dawit Wolday, Wendemagegn Embiale, Myrthe Pareyn, Geremew Tasew.

**Project administration:** Adugna Abera, Melkamu Abte, Getachew Tollera, Mesay Hailu, Geremew Tasew.

**Resources:** Adugna Abera, Henok Tadesse, Desalegn Geleta, Mahlet Belachew, Ebise Abose Djirata, Hailemariam Difabachew, Melkamu Abte, Getachew Tollera, Mesay Hailu, Johan van Griensven, Myrthe Pareyn.

**Software:** Adugna Abera, Desalegn Geleta.

**Supervision:** Adugna Abera, Dereje Beyene, Ebise Abose Djirata, Tesfahun Bishaw, Mussie Abdosh Hassen, Zalalam Olani Tugga, Endawoke Eyelachew, Henock Bekele, Zeyede Kebede, Fikre Seife, Melkamu Abte, Getachew Tollera, Mesay Hailu, Johan van Griensven, Dawit Wolday, Geremew Tasew.

**Validation:** Adugna Abera, Henok Tadesse, Dereje Beyene, Mahlet Belachew, Solomon Kinde, Hailemariam Difabachew, Dawit Wolday, Wendemagegn Embiale, Myrthe Pareyn, Geremew Tasew.

**Visualization:** Adugna Abera, Henok Tadesse, Dereje Beyene, Desalegn Geleta, Mahlet Belachew, Ebise Abose Djirata, Solomon Kinde, Hailemariam Difabachew, Abdulahi Gire, Tariku Mulatu Bore, Binyam Mohammedbirhan Berhe, Gert van der Auwera, Jean-Claude Dujardin, Wendemagegn Embiale, Myrthe Pareyn, Geremew Tasew.

**Writing – original draft:** Adugna Abera, Henok Tadesse, Melkamu Abte, Johan van Griensven, Wendemagegn Embiale, Myrthe Pareyn.

**Writing – review & editing:** Adugna Abera, Henok Tadesse, Dereje Beyene, Desalegn Geleta, Mahlet Belachew, Ebise Abose Djirata, Solomon Kinde, Hailemariam Difabachew, Tesfahun Bishaw, Mussie Abdosh Hassen, Abdulahi Gire, Tariku Mulatu Bore, Binyam Mohammedbirhan Berhe, Medhanye Habtetsion, Zalalam Olani Tugga, Endawoke Eyelachew, Worku Birhanu Sefer, Kaoutar Choukri, Jasmine Coppens, Gemechu Tadese, Kebron Haile, Henock Bekele, Zeyede Kebede, Gert van der Auwera, Fikre Seife, Getachew Tollera, Mesay Hailu, Jean-Claude Dujardin, Johan van Griensven, Dawit Wolday, Wendemagegn Embiale, Myrthe Pareyn, Geremew Tasew.

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
